



**Water Scarcity under Various Socio-economic Pathways and its Potential Effects on**
**Food Production in the Yellow River Basin**
Yuanyuan Yin [a]  Qiuhong Tang [a,*]  Xingcai Liu [a]  Xuejun Zhang [a]
[1] Key Laboratory of Water Cycle and Related Land Surface Processes, Institute of Geographic
Sciences and Natural Resources Research, Chinese Academy of Sciences, Beijing 100101, China
Correspondence to: Qiuhong Tang (tangqh@igsnrr.ac.cn)
**Abstract**: Increasing population and socio-economic development have put great pressure on water
resources of the Yellow River (YR) basin. The anticipated climate and socio-economic changes may
further increase water stress. Many studies have investigated the changes in renewable water resources
under various climate change scenarios but few have considered the joint pressure from both climate
change and socio-economic development. In this study, we assess water scarcity under various socio-
economic pathways with an emphasis on the impact of water scarcity on food production. The water
demands in the 21st century are estimated based on the newly developed Shared Socio-economic
Pathways (SSPs) and the renewable water supply is obtained from the climate projections under the
RCP 8.5 scenario. The assessment predicts that the renewable water resources and domestic water
demand are projected to first increase and then decrease, while the industrial water demand is
projected to rapidly increase in the basin during the 21st century. The water demands will put the
middle and lower reaches in conditions of severe water scarcity beginning in the next a few decades
(during 1990s-2040s). The industrial water demand is the main contributing factors to water scarcity.
The irrigation water demand is another important contributing factor under SSP3. If more than 10% of
the renewable water resources are used to sustain ecosystems, a portion of irrigated land would have
to be converted to rain-fed agriculture which would lead to a 9-38% reduction in food production.
This study highlights the links between water, food and ecosystems in a changing environment and
suggests that trade-offs should be considered when developing regional adaptation strategies.
**Key words**: water scarcity; Shared Socio-economic Pathways; climate change; Yellow River basin



1 Introduction
The Yellow River (YR) is the second-longest river in China and is regarded as the cradle of Chinese
civilization. The YR plays an important role in the development of the regional economy as the major
source of freshwater for a large amount of people living there. As of 2010, there were 113.7 million
inhabitants and 12.6 million hectares of cultivated land in the basin (YRCC, 2013). In addition, the
lower reaches of the river support the freshwater for 2.86 million hectares of irrigated area and a
population of 54.73 million located outside the basin (Fu et al., 2004). Increasing population and
socio-economic development have put great pressure on the water resources of the basin. Anticipated
climate and socio-economic changes may further increase water scarcity. The water managers of the
basin will face great challenges meeting the human and environmental requirements for water. This
water crisis in the YR basin has received much attention for many years.
Climate change and human water use are two major reasons for water crisis in the YR basin (Fu et al.,
2004; Tang et al., 2008a; Wang et al., 2012). Numerous studies have investigated the changes in water
supply due to climate change. Since the 1950s, the streamflow of the river has decreased partly
because of the decrease in precipitation and increase in temperature (Tang et al., 2008b; Xu, 2011;
Wang et al., 2012). Some recent studies showed that there has been a substantial recovery of natural
runoff over the past decade as a response to changes in precipitation, radiation and wind speed (Tang
et al., 2013; Liu et al., 2014). Climate projections suggest that temperature will continue to rise but
renewable water resources might decrease over the next few decades (Leng et al., 2015). Renewable
water resources of the YR are likely to decrease due to both precipitation decrease and temperature
increase over the next few decades (Li et al., 2012; Davie et al., 2013; Haddeland et al., 2014).
However, water resources might increase by the end of 21st century due to an increase of precipitation
(Liu et al., 2011; Leng et al., 2015). The change in water availability under climate change suggests
the need for adaptation.
Along with rapid economic development and population growth, water withdrawals from the YR
basin for industrial and household use have increased significantly. Water consumption for irrigation





has induced a streamflow decrease by about half in the past half century (Tang et al., 2007; Shi et al.,
2012). The lower reaches of the YR frequently ran dry (i.e. no streamflow in the low flow season) in
the 1980s and 1990s (Tang et al., 2008b). Thereafter, the Yellow River Conservancy Commission
(YRCC) implemented a water flow regulation rule, which enforced an upper limit on water
withdrawals for the eight provinces that rely on water supply from the river (Cai and Rosegrant, 2004).
The expected increase in economic prosperity together with a growing population, both within and
outside of the basin, will increase water demand from the river and thus water scarcity may impose
further constraints on development and social well-being (Schewe et al., 2014). As water becomes
increasingly scarce, there will be more competitions and conflicts among different water use sectors
and regions (provinces). The current water flow regulation rule, which has been enforced since the late
1990s, might not be applicable in the 21st century.
Many studies have investigated the changes in water supply under various climate change but few
have considered the joint pressure from both climate change and socio-economic development. It
becomes important to develop qualitative scenario storylines to assess future water scarcity in a
changing environment at the regional scale. These storylines would provide a grant figure of the water
use competitions among different sectors and regions and thus offer information facilitating the
development of an adaptation strategy for the river basin. A few studies have tried to describe the
main characteristics of future climate change scenarios and development pathways at the global scale
(Elliott et al., 2014; Schewe et al., 2014). These efforts, though important, are too coarse for
vulnerability assessment at the regional scale. For example, the global studies assumed an upper
availability of 40% of total annual blue water supply for human use but the human water appropriation
has been much higher than 40% in the YR basin (YRCC, 2013). Moreover, the river supplies water for
the irrigation districts in the lower reaches, which are located outside of the basin. The water demands
outside the basin are generally not considered in the global scale assessments. In this study, we present
a multi-model analysis of water supply and demand narratives under different climate change
scenarios and socio-economic pathways at the sub-basin scale (Figure 1 and Table 1). The objectives
of the analysis are: i) to describe the water supply and demand changes in a changing environment; ii)





to identify the possible time horizon when current management practices may no longer be sustainable;
iii) to investigate the contributions of different water demand sectors to water scarcity; and iv) to
assess the potential impacts of water scarcity on agricultural production.
## 2 Study area and Data
### 2.1 Study area
The YR originates in the northern foothills of the Tibetan Plateau, runs through nine provinces and
autonomous regions, and discharges into the Bohai Gulf (Figure 1). Total area of the basin is 75.2
thousand km$^2$. The YR basin lies in a temperate continental climate zone, and most parts of the basin
belong to arid or semi-arid regions. The mean temperature ranges from -5°C to 15°C in 1981-2010 in
the basin, and it increases from north to south as consequence of the decrease in latitude to the south
(Figure 2 (a)). Precipitation has large spatial variation within the whole river basin. The mean annual
precipitation ranges from 60mm to 900mm in 1981-2010, and shows an increasing trend from
northwest to southeast (Figure 2 (b)). The temperature and precipitation are projected to increase
during the 21$^{st}$ century under the RCP 8.5 emission scenario (see Figure S1 in Supplemental material).
There are six land cover types in the basin (Figure 2 (c)). The dominant land cover types are
grasslands (47.6%), croplands (26.1%), and forest and shrub-lands (13.4%). The urban and built-up
land are concentrated along the river. The croplands are mainly distributed in the lower reach of the
YR. The land-cover change influence the hydrological cycle (Tang et al., 2008b; Tang et al., 2012),
and may alter runoff (Sterling et al., 2013). However, interactions among land cover change, climate
change, and hydrological cycle are complicated. The fixed land cover map was used in this study,
which focuses on runoff responses to climatic variations.
In 2010, the population within the basin boundary was more than 100 million, representing about 9%
of China's population. The basin's GDP was represented 8% of China's GDP in 2010. Both
population and GDP are concentrated along the river (Figure 2 (d) and (e)). The projected population
increases first and then decreases during the 21$^{st}$ century (see Figure S2 (a) in Supplemental material).
The range of projected population at the end of the 21$^{st}$ century varies from 50 million to more than




100 million, with SSP5 at the bottom of the range and SSP3 at the top. The range of projected GDP at
the end of the 21$^{st}$ century varies from 21,000 billion yuan to more than 40,000 billion yuan. SSP5,
with its focus on development, has the highest GDP projections, and SSP3 representing the scenario
with lowest international co-operation has the lowest income projection (O'Neill et al., 2015).
2.2 Data
The data used in this study are summarized in Table 2. The simulated runoff data for the period 1971-
2099, and the simulated irrigation water use and crop yield data for the period 1981-2099 were
obtained from the Inter-Sectoral Impact Model Intercomparision Project (ISI-MIP) (Warszawski et al.,
2014). These model simulated data were provided at a spatial resolution of 0.5°×0.5°. The runoff data
were produced by six global gridded hydrological models (GGHMs), namely H80, MPI-HM, PRC-
GLOBWB, VIC, WaterGAP, and WBM (see Table S1 in Supplemental material). The irrigation water
use and crop yield data were produced by six global gridded crop models (GGCMs), namely EPIC,
GEPIC, LPJmL, LPJ-GUESS, pDSSAT and PEGASUS (see Table S2 in Supplemental material2).
Forcing data bias-corrected by the ISI-MIP team for the GGHMs and GGCMs were derived from
climate projections of five global climate models (GCMs), namely HadGEM2-ES, IPSL-CM5A-LR,
MIROC-ESM-CHEM, GFDL-ESM2M, and NorESM1 (see Table S3 in Supplemental material) under
the RCP 8.5 scenario (Warszawski et al., 2014). The global irrigated and rain-fed crop area data
(MIRCA2000), which consist of all major food crops such as wheat, rice, maize, and soybean, were
also obtained from ISI-MIP. The MIRCA2000 data set refers to the crop area over the period of 1998-
2002 (Portmann et al., 2010).
The gridded population and Gross Domestic Product (GDP) datasets over China were provided by the
Institute of Geographic Sciences and Resources Research (IGSRR), Chinese Academy of Sciences
(CAS). The population and GDP datasets refer to the conditions in 2005 (Fu et al., 2014; Huang et al.,
2014). The datasets were developed based on remote sensing-derived land use data and the statistical
population and GDP data of each county in China. The population and GDP data were provided with a
spatial resolution of 1 km and were resampled to 0.5° in this study with ArcGIS. The annual total
population and GDP data of China during 1981-2013 were obtained from the National Bureau of

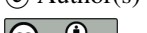



Statistics of China (NBC). Using a simple linear downscaling method (Gaffin et al., 2004), we
downscaled the annual total population and GDP data to the gridded maps. The future water demand
should be closely related to the growth of GDP and population growth in the basin, and the SSPs offer
the possibility for describing different conditions in terms of future sectoral water demand. We used
three SSPs: SSP2 (middle population and GDP growth), SSP3 (high population and low GDP growth),
SSP5 (low population and high GDP growth) (Chateau et al., 2012) in this study. Quantitative
projections for population and GDP were developed for the 2010-2099 period based on the Shared
Socioeconomic Pathways (SSP) Scenario Database data available at https:// secure.iiasa.ac.at/web-
apps/ene/SspDb. The population and GDP projections were provided at country level at five-year
intervals. The country level population data were gridded to 0.5° according to the 2010 Gridded
Population of the World (GPWv3) dataset provided by the Center for International Earth Science
Information Network (CIESIN), Columbia University. The country-level GDP data were provided in
U.S. dollars at five-year intervals. The GDP data were converted to Chinese Yuan using the official
exchange rate provided by the World Bank. The GDP data from the SSP Scenario Database were
regridded to the 0.5° GDP of China grid and were linearly interpolated in time to obtain annual values
(Gaffin et al., 2004). The assumption underlying the downscaling method is that the annual growth
rate of GDP at each grid, at any year, is equal to the growth rate of China. The domestic and industrial
water consumptions over 1997-2013 in the river basin were obtained from the China Water Resources
Bulletin (MWR, 2013). Domestic and industrial water consumptions were missing in 1998. The
observed runoff data over 1971-2000 of four major hydrologic stations at the main stream of the YR
(Lanzhou, Longmen, Sanmenxia and Huanyuankou) were collected from the Hydrological Year Book
by the Hydrological Bureau of the Ministry of Water Resources of China.
3 Method
The river basin was divided into eight sub-basins in order to understand the regional patterns of water
abundance and scarcity (Figure 1). The area of sub-basins varies from 40 to 185 thousand km$^2$ (Table
1). There are seven sub-basins along the main stream of the basin and one endorheic basin (sub-basin





VIII) that does not flow to the main stream of the river. Because the irrigation districts located outside
of the basin in lower reaches get water supply from the river, the sub-basin in the lowest reaches (sub-
basin VII) consists of one part in the river basin and these irrigation districts.
The mean annual runoff, including both the subsurface and surface runoff, is assumed to be the
renewable water resource (Oki and Kanae, 2006). In order to assess the performance of the GGHMs in
the YR basin, we compared the simulated runoff of seven GGHMs with the observed runoff at the
four selected hydrological stations (see Table S4 in Supplemental material). The simulated runoff
agrees well with the observed runoff, suggesting the GGHMs products may be used in the basin. The
GGHM simulated runoff was aggregated for each sub-basin and the river basin as a whole. In addition,
we assume only a part of the renewable water resource is available for human use. The ratio of human
water appropriation (hereafter RHWA) was about 50% during 1980s (Zhang et al., 2004) and is more
than 70% (YRCC, 2013) presently. Because the ratio largely determines water supply availability and
it may be adjusted as water stress conditions change, a range of ratio values (50, 70 and 90%) were
used in this study. It should be noted that a higher ratio means less water for environmental flow
which is detrimental to ecosystems and human society. The water flow regulation rule currently
implemented by YRCC (YRCC, 2013) sets the upper limit on water withdrawals for each sub-basin.
According to the rule, the maximum water use proportion is prescribed for each sub-basin (Table 1).
The annual water supply was calculated for each GCM-GGHM pair. There were five GCMs and six
GGHMs, making 30 model pairs. The multi-model-ensemble median of water supply from all the
available model pairs was calculated.
On the water demand side, the consumptive agricultural, domestic and industrial water demands were
considered. Agricultural water demand consists of the demands for irrigation and livestock. As the
livestock demand is relatively small and the related statistical data were unavailable in the basin
(YRCC, 2013), only irrigation demand was estimated. The GGCM estimated irrigation water demand
(IrrWD) for the major crops, namely wheat, rice, maize and soybean, was used. The IrrWD was
aggregated for each sub-basin and the river basin as a whole. The multi-model-ensemble median of
IrrWD from all the available GCM-GGCM pairs (five GCMs × six GGCMs) was calculated.



Domestic and industrial water demands were linked to the main driving forces of water in the
domestic and industrial sectors, i.e. population and GDP, respectively (Alcamo et al., 2003).
Following Alcamo et al. (2003) and Flörke et al. (2013), the water use intensity (per unit use of water)
in each sector was estimated. In the domestic sector, water use intensity should rapidly grow along
with income growth (GDP per capita). Eventually, after a maximum level is reached, water use
intensity should either stabilize or decline as income continues to grow. This process can be
represented by a sigmoid curve. Using the historical GDP per capita and domestic water user per
capita data collected in the YR basin, a sigmoid curve was established (see Figure S3(3) in
Supplemental material). In the industrial sector, the water use intensity would rapidly decrease along
with the growth in income, and eventually level off with increasing income. This process can be
represented by a hyperbolic curve. Using the historical GDP per capita and industry water user per
capita data, a hyperbolic curve was constructed for the basin (see Figure S3(b) in Supplemental
material). These curves, together with the GDP and population scenario data, were used to estimate
future domestic and industrial water demands. Technological advance, which could lead to
improvements in the efficiency of water use and a decrease in water intensity, was accounted for using
a technological change (TC) rate. In the domestic sector, TC was set as 1% per year. In the industry
sector, TC was set to 2.4% per year between 1981 and 1999, and 1% per year thereafter following
Flörke et al. (2013).
The water supply stress index (WaSSI), defined as the ratio of water demand to water supply
(McNulty et al., 2010; Averyt et al., 2013), was calculated for each sub-basin and the whole basin to
assess water abundance/scarcity condition. To investigate the contributions of different water demand
sectors to water scarcity, WaSSI was calculated for each major sector (domestic, industry and
irrigation) at the end of the 21st century. If the WaSSI  is projected to be is greater than 1, water
resources cannot sustain the socio-economic development and water scarcity occurs. The greater the
WaSSI value, the greater the water scarcity. We assume that irrigated agriculture has the lowest
priority of all water consumers under water stress. When water scarcity occurs in a given year for a
given sub-basin, irrigation was constrained by reducing the irrigated fraction of the cropland (Elliott et





al., 2014). The agricultural production of the sub-basin, calculated as calorie content of the major crop
yields, would be the sum of production over the expanded rain-fed fraction of the cropland and the
shrunken irrigated fraction. If water abundance in a given year for a given sub-basin, we assume that
no rain-fed areas were converted for irrigation.
The water supply and demands were assessed for each year but the 30-year moving averages during
1981-2099 were computed and illustrated. The 30-year window ensures that year-to-year variability
dose not dominate the signal. The center year of the 30-year moving average was used to denote the
30-year period. For example, the average of the historical period of 1981-2010 was denoted as 1995.
4 Results
**4**.1 Changes of supply water
Figure 3 shows the supply water during 1995-2084 in the YR basin and eight sub-basins under three
different ratios of human water appropriation (RHWA). With the increase of RHWA, the supply water
is projected to increase during the 21$^{st}$ century in the YR basin. The average supply water is 34.8, 48.8
and 62.7 billion m$^3$ per year during the historical period under three RHWA -- 50, 70 and 90%,
respectively. The supply water is projected to decrease from 1995-2058 due to the increase of air
temperature (see Figure S1 in Supplemental material), and is projected to decrease from 2059-2084
due to the increase of precipitation under all RHWAs (see Figure S1 in Supplemental material). The
result is consistent with the conclusions from Zhao et al. (2009). The supply water is projected to be
36, 51 and 65 billion m$^3$ per year at the end of the 21$^{st}$ century under RHWA50, RHW70 and
RHWA90, respectively, with increasing by about 4% compared with the supply water during the
historical period. The supply water is also projected to first decrease and then increase, and the supply
water is also projected to increase with the increase of RHWA in each sub-basin during the 21$^{st}$
century. The average supply water of sub-basin III has the maximum value of 13 billion m$^3$ per year
during the historical period and rises to 23.5 billion m$^3$ per year by the end of the 21$^{st}$ century under
RHWA50. The average supply water of sub-basin VIII has the minimum value of 0.09 billion m$^3$ per



year during the historical period and rises to 0.16 billion $m^3$ per year by the end of the $21^{st}$ century
under RHWA50.
4.2 Changes of total and sectoral water demand
Figure 4 and 5 show the estimated total and sectoral (domestic, industrial and irrigation) water demand
in the YR basin and eight sub-basins under three SSPs from 1995 to 2085. In the YR basin, the total
water demand is projected to increase from 27.8 billion $m^3$ $yr^{-1}$ in 1995 to close to 55, 44 and 69
billion $m^3$ $yr^{-1}$ in 2084 under SSP2, SSP3 and SSP5, respectively. This increase is primarily driven by
the growth in the industrial water demand, accounting for large than 53% of the total in 2084.
Irrigation is the dominant water use sector during the period 1995-2035, but industry is the dominant
water use sector during the period 2036-2084. Domestic water demand accounts for less than 13%,
and is projected to increase and then decrease during 1995-2084. The domestic water demand is
projected to change from 3 billion $m^3$ $yr^{-1}$ in 1995 to 2.8, 3.6, and 2.3 billion $m^3$ $yr^{-1}$ in 2084 under
SSP2, SSP3 and SSP5, respectively. The industrial water demand is projected to increase from 4.6
billion $m^3$ $yr^{-1}$ in 1995 to about 35, 23, and 50 billion $m^3$ $yr^{-1}$ in 2084 under SSP2, SSP3 and SSP5,
respectively. Industrial water demand is projected to rapidly increasing during the $21^{st}$ century. The
rate of industrial water demand is about 4.3, 2.4 and 6.2 billion $m^3$ per ten years. The irrigation water
demand is projected to increase from 20 billion $m^3$ $yr^{-1}$ in 1995 to close to 17 billion $m^3$ $yr^{-1}$ in 2084
under RCP 8.5. Irrigation water demand is projected to not increase substantially during 1995-2030,
and is projected to decrease during 2031-2084, with decreasing by close to 16% compared with the
irrigation water in 1995. The total water demand and industrial water demand is also projected to
increase, and the domestic water demand is also projected to increase and then decrease in each sub-
basin during the $21^{st}$ century under all SSPs. In the sub-basin III, although the industrial water demand
would increase rapidly, irrigation is always the dominant water use sector during the $21^{st}$ century. In
the sub-basin I, II, IV, and VI, the industry is always the dominant water use sector during the $21^{st}$
century.





### 4.3 Water abundance/scarcity and sectoral contributions to water scarcity

Figure 6 shows the average annual WaSSI for the YR basin and eight sub-basins throughout the 21$^{st}$ century under three different RHWAs and three different SSPs. The WaSSI is projected to increase due to the water demand increase during the 21$^{st}$ century. Under RHWA50, the YR basin is projected to have a WaSSI greater than 1 after 2010s for SSP2 and SSP3 as well as after 2000s for SSP5, meaning than water demand outstrip supply water. The WaSSI is projected to decrease with the increase of RHWA. Under RHWA90, the water scarcity would only occur after 2050 for SSP5. The upper reaches of the YR basin (sub-basins I, II, and III) are projected to have a WaSSI less than 1, meaning that the water would be abundant, during the 21$^{st}$ century for all SSPs under all RHWAs. The endorheic basin of the YR basin (sub-basin VIII) is the only region in which the WaSSI is always larger than 1, meaning that the water would be scarce, during the 21$^{st}$ century for all SSPs under all RHWAs. In the middle and lower reaches of the YR basin (sub-basins IV, V, VI, and VII), the WaSSI would begin to be large than 1 at the beginning of the 21$^{st}$ century under RHWA50. With the increase of RHWA, a water resource scarcity would begin to occur later. When the RHWA reaches to 90%, the water would be abundant during 1995-2084 in sub-basins IV and VII under SSP2 and SSP3.

Figures 7 shows the WaSSI for total water demand and for sectoral (domestic, industrial and irrigation) water demands for the YR basin and eight sub-basins at the end of the 21$^{st}$ century under three different SSPs under RHWA50. In the YR basin, the WaSSI for total water demand is large than 1 under each SSP, meaning that the water would be scarce at the end of the 21$^{st}$ century. Among the three different water demand sectors, industrial sector is projected to have the largest WaSSI large than 1 for SSP2 and SSP5, and domestic sector is projected to have the smallest WaSSI less than 0.1. The WaSSI based only agricultural demand is about 0.5. With the increase of RHWA, the WaSSI for each water demand sectors would decrease (see Figure S4 and S5 in Supplemental material). A large amount of GDP would lead the industrial water demand to be the main contributing factor to WaSSI for all sub-basins except sub-basin III. In sub-basin III, the irrigation water demand is the main contributing factor to WaSSI, and the industrial water demand is another important contributing factor.





Because both population and GDP are concentrated in the middle and lower reaches of the basin, the
WaSSI for those sub-basins is larger than one for the sub-basins located in the upper reaches.
4.4 Agricultural loss due to irrigation water scarcity
The climate change and the scarcity of water available for irrigation in the YR basin would have
significant implications for the food security of these regions. Considering the $CO_2$ fertilization effect,
the agricultural production would be enhanced by climate change, and is projected to increase by close
to 23% compared with the production during the historical period in the YR basin at the end of the
century (Figure 8). Irrigation water scarcity could necessitate the reversion of cropland from irrigated
to rain-fed management, and would lead to decreased agricultural production. Under RHWA50,
irrigation water scarcity in the basin could necessitate the reversion of all cropland from irrigated to
rain-fed management under SSP2 and SSP5, and the reversion of 45 thousand $km^2$ of cropland from
irrigated to rain-fed management under SSP3 by the end-of-$21^{st}$-century. Considering the $CO_2$
fertilization effect, irrigation water scarcity would lead to 38% of present-day total production
reduction under SSP2 and SSP5, and 21% of present-day total production reduction under SSP3 in
2084 (Figure 8). The change rate of production is projected to decrease with the increase of RHWA.
Under RHWA90, the reduction of agriculture production (close to 10%) in 2084 only occurs under
SSP5. Considering the climate and water supply stress impact, the reduction of agriculture production
in 2084 is about 10% for SSP2 and SSP5 under RHWA50 as well as SSP5 under RHWA70. Under
RHWA90, the agriculture production is projected to increase under each SSP at the end of the $21^{st}$
century.
5 Discussion
The renewable water resource will be affected by projected changes in precipitation and temperature
(Schewe et al., 2014), and the RHWA. The water supply in the YR basin would first decrease and then
increase in varying degrees due to the impact of temperature and precipitation rise over the $21^{st}$
century (see Figure S1 in Supplemental material). However, the true water shortage might be larger
because the CMIP5 models may overestimate the magnitude of precipitation in the YR basin during



the 21st century (Chen and Frauenfeld, 2014). The RHWA of the YR basin has increased to 75.6%
during the beginning of the 21st century (Shi et al., 2012) from about 50% in 1980s (Zhang et al.,
2004). The increase in RHWA tends to result in an increase in water supply, and results in a reduction
in irrigation water scarcity and a loss of agriculture production (Figure 1and Figure 8). Therefore,
improvement of the RHWA could alleviate the water shortages in this region. However, because of
the different geographical and economic conditions among the sub-basins, the impact of the RHWA
should be considered when we analyze the water resource of the sub-basins.
To quantify domestic and industrial water demand are complicated because the future of the water
demand will be influenced by a combination of social, economic, and political factors. However, a
few of the hydrologic modeling frameworks have associated methods to estimate water demand, e.g.
H08 (Hanasaki et al., 2010; Hanasaki et al., 2013a and 2013b), PCR-GLOBWB (Wada et al., 2011;
Wada et al., 2014), WaterGAP (Flörke et al., 2013). The differences in these approaches result in
significantly different projections even with same set of scenario assumptions (Wada et al., 2016).
Wada's research showed that in China, WaterGAP used in this study projects a much larger industrial
water demand than H08 and PCR-GLOBWB (Wada et al., 2016). So our study might overestimate the
water scarcity and the contribution of industrial water demand to water scarcity.
The decrease in runoff and the increase in domestic and industrial water use are the main factors
leading to the water resource crisis from 1995-2020, while the increase in industrial water use is the
main factor leading to the water resource crisis after 2020 in the YR basin and the sub-basins located
in the middle and lower reaches. The structural changes in water intensity for both domestic and
industrial use are associated with living standards and levels of industrialization (Alcamo et al., 2003;
Flörke et al., 2013). In this study, we assumed that the structural changes in water intensity for
domestic and industrial use in the eight sub-basins were the same. This assumption might lead to an
overestimate of the domestic and industrial water use in the middle and lower reaches and to an
underestimate of the domestic and industrial water use in the upper reaches. Therefore, the difference
of the structural changes in water use intensity should be considered when we analyze the water
resource of the sub-basins.





With the currently implemented water flow regulation rule, water is projected to be scarce in sub-
basins located the middle and lower reaches of the YR basin characterized by a generally large
population and GDP, while water is projected to be abundant in sub-basins located in the upper
reaches of the YR basin characterized by a small population and GDP during the 21st century. In order
to alleviate the water shortages in the middle and lower reaches, a new water flow regulation rule
could be adopted.
In order to solve the problem of water resource shortages in the more arid and industrialized north of
China, the South-to-North Water Diversion Project has been undertaken. One aim of the project is to
channel the fresh water from the Yangtze River in southern China to the YR basin (YRCC, 2013). By
2030, about 9.7 billion m³ of fresh water from the Yangtze River would be drawn to the YR basin
(YRCC, 2013). This could alleviate the water shortage in the YR basin to some degree.
6 Conclusions
In this study, we assessed the change in renewable water resource of the YR basin under climate
change and the changes in domestic and industrial water demand in the basin under socio-economic
change in the 21st century. The results show that the renewable water resources are projected to first
increase and then decrease in the YR basin and each sub-basin with the increase of temperature and
precipitation under RCP 8.5 in the 21st century. Irrigation is the dominant water use sector during the
period 1995-2035, but industry is the dominant water use sector during the period 2036-2084. With
social and economic development, domestic water demand is projected to increase and then decrease,
and industrial water demand is projected to rapidly increasing during the 21st century.
Water is always scarce in the endorheic basin, while water is always abundant in the sub-basins
located in the upper reaches of the YR basin in the 21st century under all RHWAs and SSPs. Due to
water demand increase in industrial sectors, the available water resources cannot sustain all the water
use sectors beginning in the next a few decades in the YR basin and the sub-basins located in the
middle and lower reaches of the basin. The water resource shortage is most serious under the
conventional development scenario (SSP5) and 90% of the renewable water resources cannot sustain





367 all the water use sectors in the YR basin. With the three water demand sectors considered, the

368 industrial water demand is the main contributing factors to water scarcity. The irrigation water

369 demand is another important contributing factor under SSP3.

370 Although climate change may have a positive impact on agriculture through the $CO_2$ fertilization

371 effect in most regions of the YR basin (Yin et al., 2015), irrigation water scarcity would lead to the net

372 loss of agricultural production. With the $CO_2$ fertilization effect, the irrigation water scarcity in the YR

373 basin could necessitate losses of production (more than about 10% of present-day total). However, the

374 difference of the structural changes in water use intensity and the difference of RHWA have not been

375 considered in this study. This might lead to an overestimate of the water abundance and scarcity.

376 Nevertheless, this study highlights the linkage between water and food security in a changing

377 environment in the YR basin, and suggests that the trade-off should be considered when developing

378 regional adaptation strategies.

379 Reference

380 Alcamo, J., Döll, P., Henrichs, T., Kaspar, F., Lehner, B., Rösch, T., and Siebert, S., 2003. Development and testing of

381 the WaterGAP2 global model of water use and availability. Hydrological Sciences Journal, 48(3), 317-337, doi:

382 10.1623/hysj.48.3.317.45290

383 Averyt, A., Meldrun, J., Caldwell, P., Sun, G., McNulty, S., Huber-Lee, A., and Madden, N., 2013. Sectoral

384 contributions to surface water stress in the coterminous United States. Environmental Research Letters, 8,

385 doi:10.1088/1748-9326/8/3/035046

386 Cai, X., and Rosegrant, M. W., 2004. Optional water development strategies for the Yellow River basin: Balancing

387 agricultural and ecological water demands. Water Resources Research, 40, W08S04, doi: 10.1029/2003WR002488

388 Chateau, J., Dellink, R., Lanzi, E., and Magné, B., 2012. Long-term economic growth and environmental pressure:

389 Reference scenarios for future global projections. OECD Working Paper, ENV/EPOC/WPCID (2012) 6

390 Chen, L., and Frauenfeld, O. W., 2014. A comprehensive evaluation of precipitation simulations over China based on

391 CMIP5 multimodel ensemble projections. Journal of Geophysical Research: Atmospheres, 119, 5767-5786, doi:

392 10.1002/2013JD021190

393 Davie, J. C. S., Falloon, P. D., Kahana, R., Dankers, R., Betts, R., Portmann, F. T., Wisser, D., Clark, D. B., Ito, A.,

394 Masaki, Y., Nishina, K., Fekete, B., Tessler, Z., Wada, Y., Liu, X., Tang, Q., Hagemann, S., Stacke, T., Pavlick, R.,





Schaphoff, S., Gosling, S. N., Franssen, W., and Arnell. N., 2013. Comparing projections of future changes in runoff
from hydrological and biome models in ISI-MIP. Earth System Dynamics, 4, 359-374, doi: 10.5194/esd-4-359-2013
Elliott, J., Deryng, D., Müller, C., Frieler, K., Konzmann, M, Gerten, D., Glotter, M., Flörke, M., Wada, Y., Best, N.,
Eisner, S., Fekete, B. M., Folberth, C., Foster, I., Gosling, S. N., Haddeland, I., Khabarov, N., Ludwing, F., Masaki, Y.,
Olin, S., Rosenzweig, C., Ruane, A. C., Satoh, Y., Schmid, E., Stacke, T., Tang, Q., and Wisser, D., 2014. Constraints
and potentials of future irrigation water availability on agricultural production under climate change. Proceedings of the
National Academy of Sciences of the United States of America, 111(9), 3239-3244, doi: 10.1073/pnas.1222474110
Flörke M., Kynast E., Bärlund I., Eisner S., Wimmer F., and Alcamo J., 2013. Domestic and industrial water uses of the
past 60 years as a mirror of socio-economic development: A global simulation study. Global Environment Change, 23,
144-156, doi: 10.1016/j.gloenvcha.2012.10.018
Fu, G. B., Chen, S. L., Liu, C. M., and Shepard, D., 2004. Hydro-climatic trends of the Yellow River basin for the last
50 years. Climatic Change, 65, 149-178, doi: 10.1023/B:CLIM.0000037491.953.95.bb
Fu, J. Y., Jiang, D., and Huang, Y. H., 2014. 1 km grid population dataset of China (2005, 2010). Acta Geographic
Sinica, 69 (Supplement), 136-139, doi: 10.3974/geodb.2014.01.06.V1
Gaffin, S. R., Rosenzweig, C., Xing, X. S., and Yetman, G., 2004. Downscaling and geo-spatial gridding of socio-
economic projections from the IPCC Special Report on Emissions Scenarios (SRES). Global Environmental Change,
14, 105-123, doi: 10.1016/j.gloenvcha.2004.02.004
Haddeland, I., Heinke, J., Biemans, H., Eisner, S., Flörke, M., Hanasaki, N., Konzmann, M., Ludwig, F., Masaki, Y.,
Schewe, J., Stacke, T., Tessler, Z. D., Wada, Y., and Wisser, D., 2014. Global water resources affected by human
interventions and climate change. Proceedings of the National Academy of Sciences of the United States of America,
111(9), 3251-3256, doi: 10.1073/pnas.1222475110
Hanasaki, N., Fujimori, S., Yamamoto, T., Yoshikawa, S., Masaki, Y., Hijioka, Y., Kainuma, M., Kanamori, Y., Masui,
T., Takahashi, K., and Kanae, S., 2013a. A global water scarcity assessment under Shared Socio-economic Pathways –
Part 1: Water use. Hydrology and Earth System Sciences, 17, 2375-2391, doi:10.5194/hess-17-2375-2013
Hanasaki, N., Fujimori, S., Yamamoto, T., Yoshikawa, S., Masaki, Y., Hijioka, Y., Kainuma, M., Kanamori, Y., Masui,
T., Takahashi, K., and Kanae, S., 2013b. A global water scarcity assessment under Shared Socio-economic Pathways –
Part 2: Water availability and scarcity. Hydrology and Earth System Sciences, 17, 2393-2413, doi:10.5194/hess-17-

422   2393-2013

Hanasaki, N., Inuzuka, T., Kanae, S., and Okim T., 2010. An estimation of global virtual water flow and sources of
water withdrawal for major crops and livestock products using a global hydrological model. Journal of Hydrology, 382:
232-244, doi:10.1016/j.jhydrol.2009.09.028





Huang, Y. H., Jiang, D. and Fu, J. Y., 2014. 1 km grid GDP data of China (2005, 2010). Acta Geographic Sinica, 69
(Supplement), 140-143, doi: 10.3974/geodb.2014.01.07.V1
Leng, G., Tang, Q., Huang, M., Hong, Y., and Ruby, L., 2015. Projected changes in mean and interannual variability of
surface water over continental China. Science China: Earth Sciences, 58(5), 739-754, doi: 10.1007/s11430-014-4987-0
Li, L., Shen, H. Y., Dai, S., Xiao, J. S., and Shi, X. H., 2012. Response of runoff to climate change and its future
tendency in the source region of Yellow River. Journal of Geographical Sciences, 23(3), 431-440, doi: 10.1007/s11442-
012-0937-y
Liu, L. L., Liu, Z. F., Ren, X. Y., Fischer, T., and Xu, Y., 2011. Hydrological impacts of climate change in the Yellow
River Basin for the 21st century using hydrological model and statistical downscaling model. Quaternary International,
244, 211-220, doi: 10.1016/j.quaint.2010.12.001
Liu, X., Zhang, X. J., Tang, Q., and Zhang, X. Z., 2014. Effects of surface wind speed decline on modeled hydrological
conditions in China. Hydrology and Earth System Sciences, 18, 2803-2813, doi: 10.5194/hess-18-2803-2014
McNulty, S., Sun, G., Myers, J., Cohen, E., and Caldwell, P., 2011. Robbing peter to pay paul: Tradeoffs between
ecosystem carbon sequestration and water yield. In Potter, K.W., and D.K. Frevert (Eds). Watershed Management 2010:
Innovations in Watershed Management under Land Use and Climate Change. Reston, VA: American Society of Civil
Engineers, 2011
MWR (Ministry of Water Resources of the People's Republic of China). China Water Resources Bulletin (2013).
Beijing: China Water & Power Press (in Chinese)
O'Neill, B. C., Kriegler, E., Ebi, K. L., Kemp-Benedict, E., Riahi, K., Rothman, D. S., van Ruijven, B. J., van Vuuren,
D. P., Birkmann, J., Kok, K., Levy, M., and Solecki, W., 2015. The roads ahead: Narratives for shared socioeconomic
pathways describing world futures in the 21st century. Global Environmental Change,
doi:10.1016/j.gloenvcha.2015.01.004
Oki, T., and Kanae, S., 2006. Global hydrological cycles and world water resources. Science, 313(5790), 1068-1072,
doi: 10.1126/science.1128845
Portmann, F. T., Siebert, S., and Döll, P., 2010. MIRCA2000 – Global monthly irrigated and rain-fed crop areas around
the year 2000: A new high-resolution data set for agricultural and hydro- logical modeling. Global Biogeochemical
Cycles, 24, 1-24, doi: 10.1029/2008GB003435
Schewe, J., Heike, J., Gerten, D., Haddeland, I., Arnell, N. W., Clark, D. B., Dankers, R., Eisner, S., Fekete, B. M.,
Colón-González F. J., Gosling, S. N., Kim, H., Liu, X., Masaki, Y., Portmann, F. T., Satoh, Y., Stacke, T., Tang, Q.,
Wada, Y., Wisser, D., Albrecht, T., Frieler, K., Piontek, F., Warszawski, L., and Kabat, P., 2014. Multimodel





assessment of water scarcity under climate change. Proceedings of the National Academy of Sciences of the United
States of America, 111, 3245-3250, doi: 10.1073/pnas.1222460110
Shi, C. X., Zhou, Y. Y., Fan, X. L., and Shao, W. W., 2012. A study on the annual runoff change and its relationship
with water and soil conservation practices and climate change in the middle Yellow River basin. Catena, 100, 31-41,
doi: 10.1016/j.catena.2012.08.007
Sterling, S. M., Ducharne, A., and Polcher J., 2013. The impact of global land-cover change on the terrestrial water
cycle. Nature Climate Change, 3(4): 385-390, doi:10.1038/NCLIMATE1690
Tang, Q., Oki, T., Kanae, S., and Hu, H., 2007. The influence of precipitation variability and partial irrigation within
grid cells on a hydrological simulation. Journal of Hydrometeorology, 8, 499-512, doi: 10.1175/JHM589.1
Tang, Q., Oki, T., Kanae, S., and Hu, H., 2008a. Hydrological cycles change in the Yellow River basin during the last
half of the twentieth century. Journal of Climate, 21, 1790-1806, doi: 10.1175/2007JCLI1854.1
Tang, Q., Oki, T., Kanae, S., and Hu, H., 2008b. A spatial analysis of hydro-climatic and vegetation condition trends in
the Yellow River basin. Hydrological processes, 22, 451-458, doi: 10.1002/hyp.6624
Tang, Q., Vivoni, E. R., Muñoz-Arriola, F., and Lettenmaier, D. P., 2012. Predictability of evapotranspiration patterns
using remotely sensed vegetation dynamics during the North American monsoon. Journal of Hydrometeorology, 13,
103-121, doi:10.1175/JHM-D-11-032.1
Tang, Y., Tang, Q., Tian, F., Zhang, Z., and Liu, G., 2013. Responses of natural runoff to recent climatic variations in
the Yellow River basin, China. Hydrology and Earth System Sciences, 17, 4471-4480, doi: 10.5194/hess-17-4471-2013
Wada, Y., Flörke, M., Hanasaki, N., Eisner, S., Fischer, G., Tramberend, S., Satoh, Y., van Vliet, M. T. H., Yillia, P.,
Ringler, C., Burek, P., and Wiberg, D., 2016. Modeling global water use for the 21st century: The Water Futures and
Solutions (WFaS) initiative and its approaches. Geoscientific Model Development, 9, 175-222, doi:10.5194/gmd-9-

477    175-2016

Wada, Y., van Beek, L. P. H., Viviroli, D., Dürr, H. H., Weingartner, R., and Bierkens, M. F. P., 2011. Global monthly
water stress 2: Water demand and severity of water stress. Water Resources Research, 47, W07518,
doi:10.1029/2010WR009792
Wada, Y., Wisser, D., and Bierkens, M. F. P., 2014. Global modeling of withdrawal, allocation and consumptive use of
surface water and groundwater resources. Earth System Dynamics, 5, 15-40, doi:10.5194/esd-5-15-2014
Wang, S. J., Yan, M., Yan, Y. X., Shi, C. X., and He, L., 2012. Contributions of climate change and human activities to
the changes in runoff increment in different sections of the Yellow River. Quaternary International, 282, 66-77, doi:
10.1016/j.quaint.2012.07.011





Warszawski, L., Frieler, K., Huber, V., Piontek, F., Serdeczny, O., and Schewe, J., 2014. The Inter-Sectoral Impact
Model Intercomparison Projection (ISI-MIP): Project framework. Proceedings of the National Academy of Sciences of
the United States of America, 111, 3228-3232, doi: 10.1073/pnas.1312330110
Xu, J., 2011. Variation in annual runoff of the Wudinghe River as influenced by climate change and human activity.
Quaternary International, 244, 230-237, doi: 10.1016/j.quaint.2010.09.014
YRCC (Yellow River Conservancy Commission), 2013. Comprehensive planning of Yellow River Basin (2012-2030).
Zhengzhou: The Yellow River Water Conservancy Press (in Chinese)
Yin, Y., Tang, Q., and Liu, X., 2015. A multi-model analysis of change in potential yield of major crops in China under
climate change. Earth System Dynamics, 6, 45-59, doi: 10.5194/esd-6-45-2015
Zhang, H. M., Niu, Y. G., Wang, B. X., and Li, S. M., 2004. The Yellow River water resources problems and
countermeasures. Hydrology, 24(4), 26-31 (in Chinese)
Zhao, F. F., Xu, Z. X., Zhang, L., and Zou, D. P., 2009. Streamflow response to climate variability and human activities
in the upper catchment of the Yellow River Basin. Science in China Series E: Technological Sciences, 52(11), 3249-
3256, doi: 10.1007/s11431-009-0354-3





Table captions
Table 1 The eight sub-basins of the Yellow River (YR) basin.

| Sub-basins | | Area ($\times 10^3$ km$^2$) | Water use pro-portion (%) | Irrigated area (km$^2$) | Rain-fed area (km$^2$) | Note |
|---|---|---|---|---|---|---|
| Upper reaches | I | 127 | 0.57 | 219 | 27 | Above Longyangxia station |
| | II | 87 | 8.23 | 2, 680 | 1, 706 | LYX to LZ |
| | III | 157 | 37.45 | 23, 692 | 2, 106 | LZ to HKZ |
| Middle reaches | IV | 107 | 3.5 | 1, 591 | 3, 940 | HKZ to LM |
| | V | 185 | 16.64 | 25, 422 | 12, 311 | LM to SMX |
| | VI | 40 | 6.16 | 5, 717 | 2, 956 | SMX to HYK |
| Lower reaches | VII | 50.6 | 27.2 | 42, 824 | 2, 430 | HYK to LJ, and irrigation districts outside the basin |
| Endorheic basin | VIII | 42 | 0.25 | 446 | 225 | Endorheic basin |

Note: LYX (Longyangxia), LZ (Lanzhou), HKZ (Hekouzhen), LM (Longmen), SMX (Sanmenxia), HYK
(Huanyuankou), LJ (Lijin), YL basin (Yellow River basin). Sub-basin consists partly of the river basin but
also includes irrigation districts outside the basin that have water supplied by the river.



Table 2 Datasets used in this study.

| Datasets | | Spatial and temporal resolution | Source |
|---|---|---|---|
| Simulated runoff data | | 0.5°×0.5°; 1971-2099 | The Inter-Sectoral Impact Model Intercomparison Project (ISI-MIP) |
| Simulated yield data | | 0.5°×0.5°; 1971-2099 | |
| Simulated irrigation water data | | 0.5°×0.5°; 1971-2099 | |
| Rain-fed and irrigation area data | | 0.5°×0.5°; 2000 | |
| Popu-lation data | 1 km grid population dataset of China | 1km×1km; 2005 | Institute of Geographic Sciences and Natural Resources Research |
| | Historical population data of China | country; 1981-2013 | National Bureau of Statistics of China |
| | SSP population data[a] | 0.5°×0.5°; 2010-2099 | ISI-MIP |
| GDP data | 1 km grid GDP dataset of China | 1km×1km; 2005 | Institute of Geographic Sciences and Natural Resources Research |
| | Historical GDP data of China | country; 1981-2013 | National Bureau of Statistics of China |
| | SSP GDP data[a] | country; 2010-2099 | Organization for Economic Co-operation and Development (OECD) |
| Official exchange rate data | | country; 2005 | World bank |
| Domestic water consumption data of Yellow River Basin | | Yellow River basin; 1997, 1999-2013 | China Water Resources Bulletin (1997, 1999-2013) |
| Manufacture water consumption data of Yellow River Basin | | Yellow River basin; 1997, 1999-2013 | |
| Observed runoff data | | 1971-2000 | Hydrological Bureau of the Ministry of Water Resources of China |

Note: a SSP is short for Shared Socioeconomic Pathways.



Figure captions

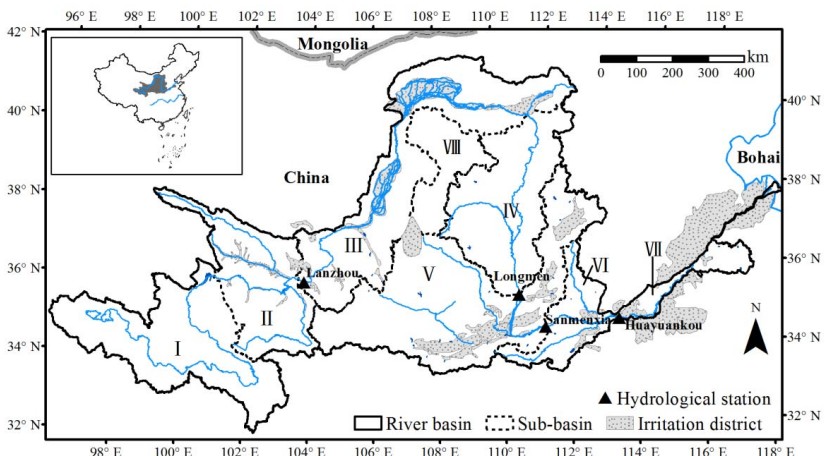


Figure 1 The Yellow River (YR) basin and the eight sub-basins, and location of the
hydrological stations used in this study.





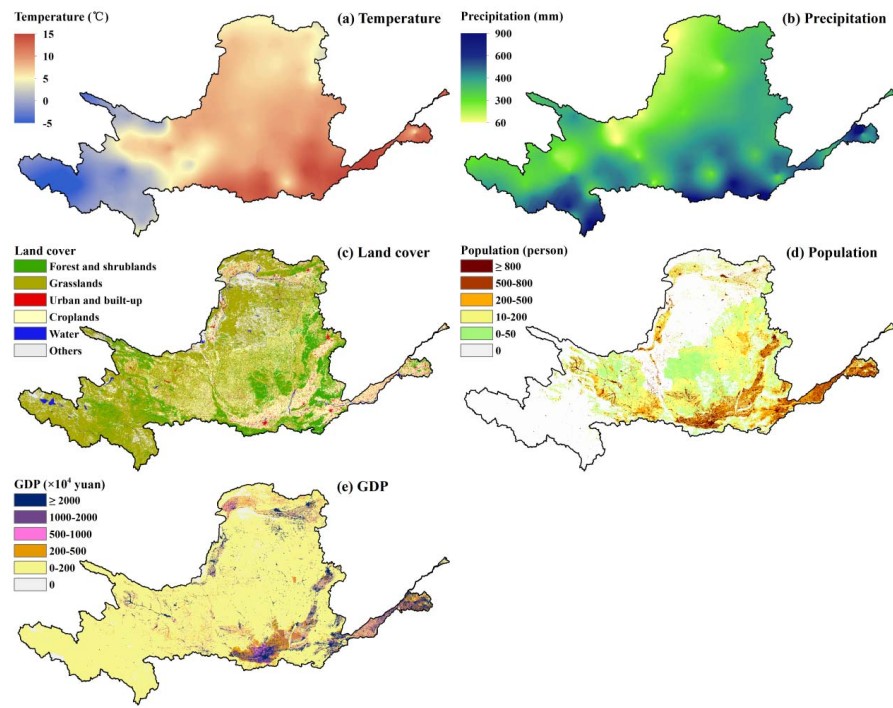


Figure 2 Maps of (a) mean temperature (1981-2010), (b) annual mean precipitation (1981-2010), (c) land cover in 2010, (d) population in 2010, and (e) gross domestic product (GDP) in 2010 in the Yellow River (YR) Basin.





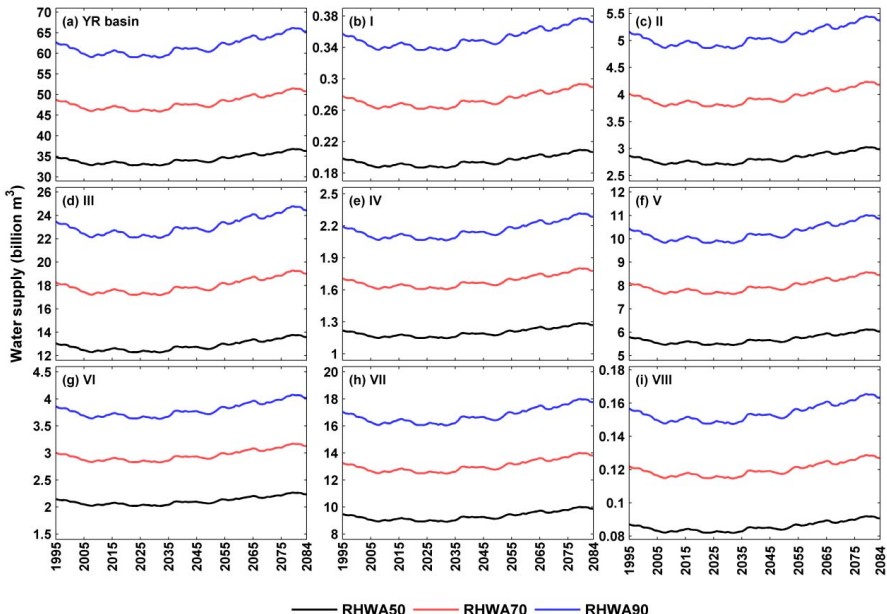


Figure 3 Supply water during 1995-2084 in the Yellow River (YR) basin and eight sub-basins

under three different ratios of human water appropriation (RHWA). The three ratios of human

water appropriation are 50%, 70% and 90%, respectively.




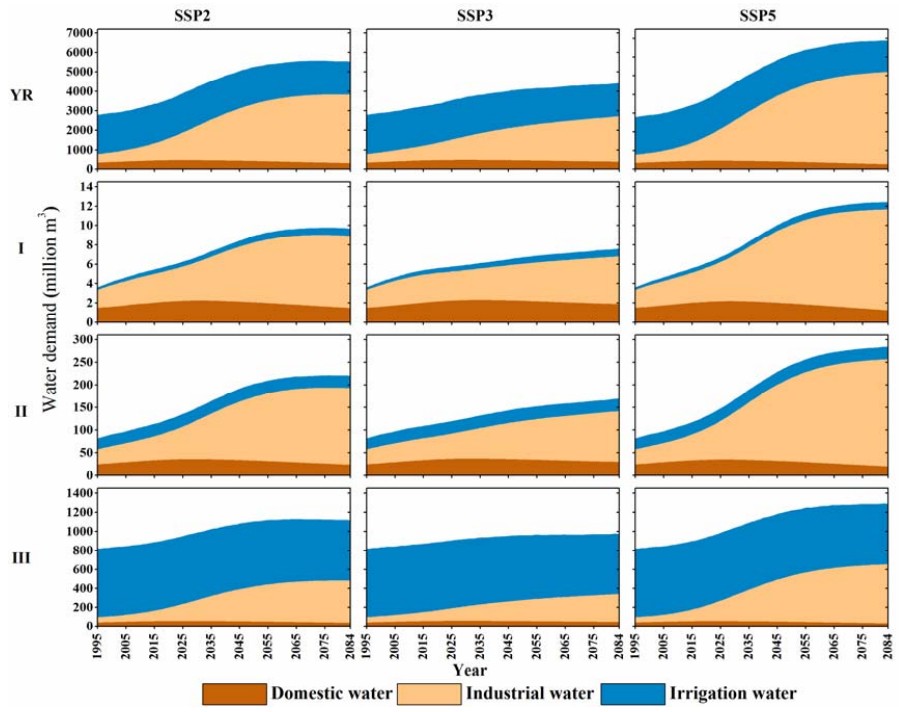


Figure 4 Estimated sectoral (domestic, industrial and irrigation) and total water demand in

Yellow River (YR) basin and sub-basins in the upper reaches from 1995 to 2085 in million

$m^3$ $yr^{-1}$.





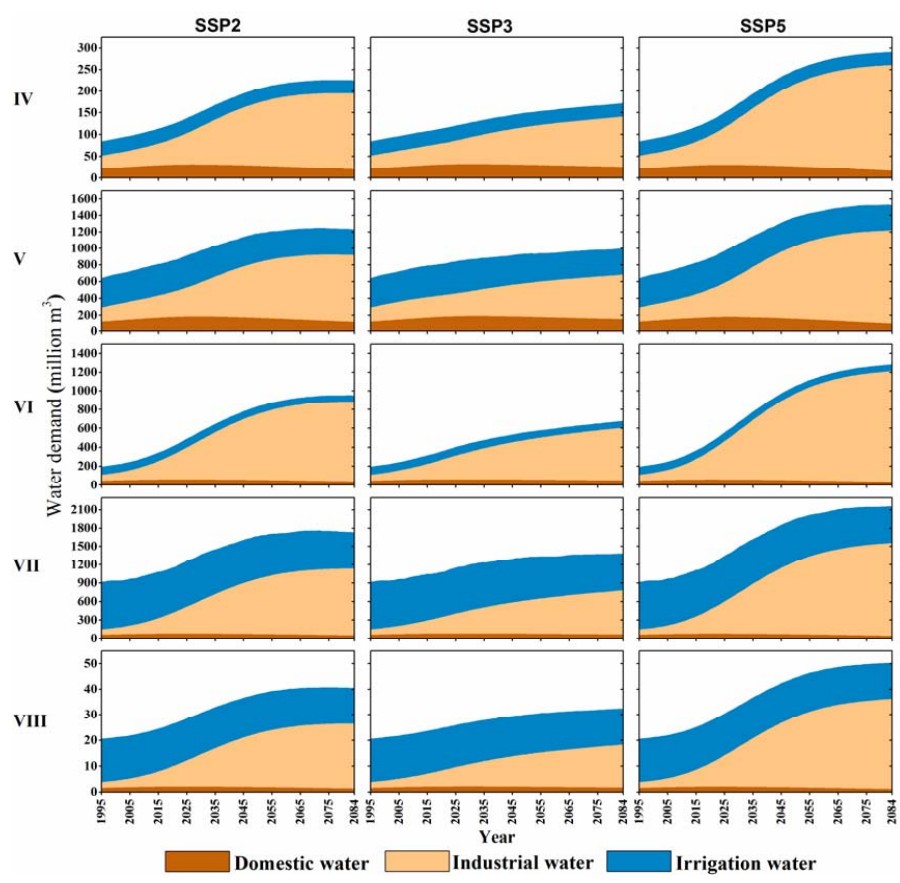


Figure 5 Estimated sectoral (domestic, industrial and irrigation) and total water demand in

sub-basins in the middle and lower reaches from 1995 to 2085 in million $m^3$ $yr^{-1}$.

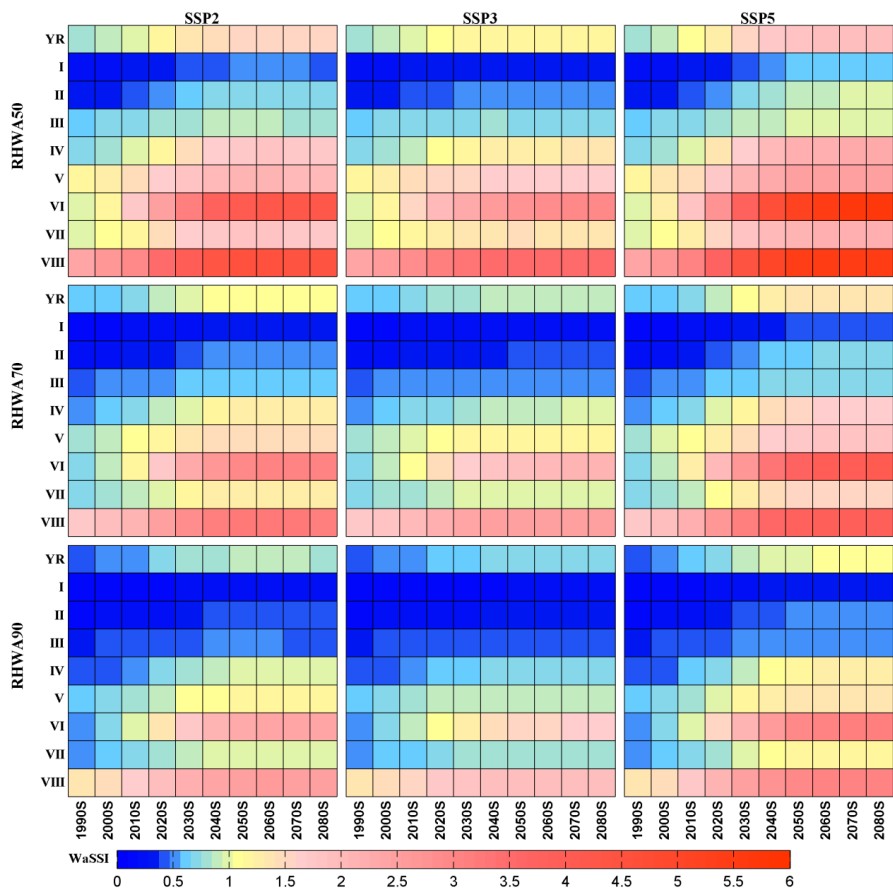


Figure 6 Average annual water supply stress index (WaSSI) for the Yellow River (YR) basin
and eight sub-basins throughout the 21st century under three different ratios of human water
appropriation (RHWA) and three different Shared Socio-Economic Pathways (SSPs). The
WaSSI are calculated for each decade. The water stress occurs in a given basin when WaSSI
is greater than 1.





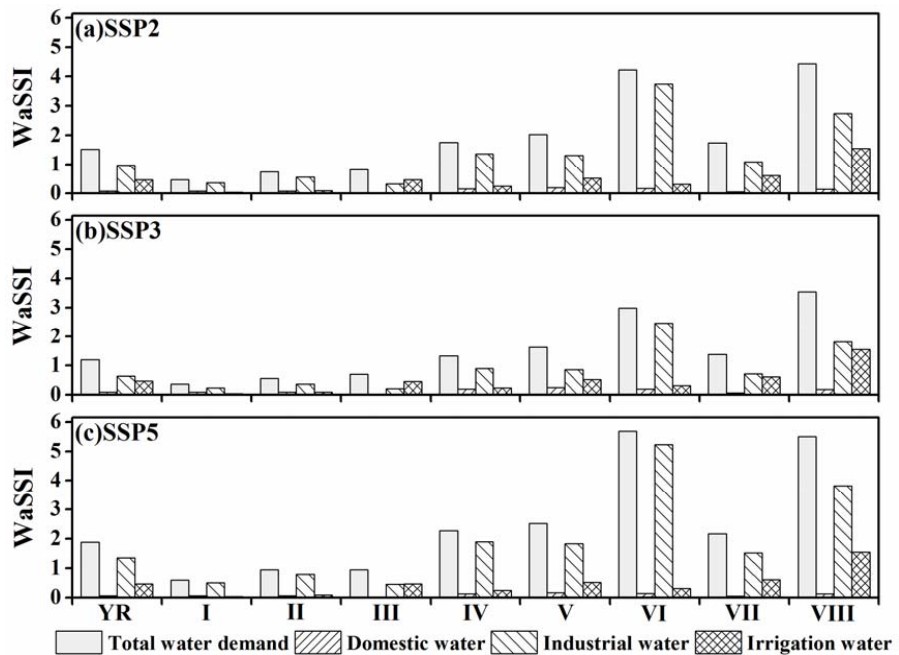


Figure 7 WaSSI for total water demand and for sectoral (domestic, industrial and irrigation)
water demands for the Yellow River (YR) basin and eight sub-basins at the end of the 21st
century under three different Shared Socio-Economic Pathways (SSPs) under RHWA50.





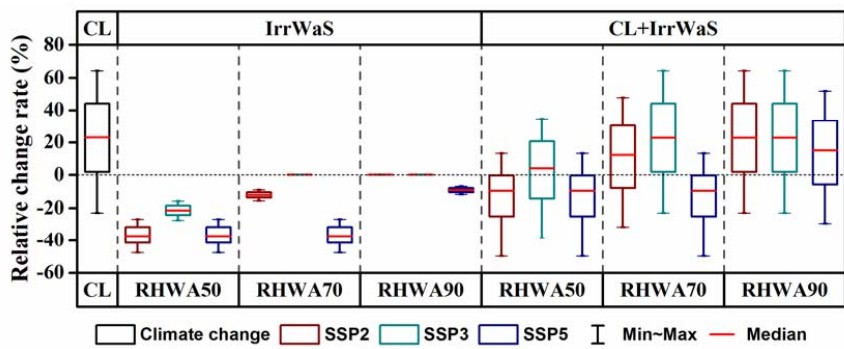


Figure 8 Comparison of relative change rate of agriculture production for only climate impact
(CL), only irrigation water scarcity impact (IrrWaS), and climate and irrigation water scarcity
impact (CL+ IrrWaS) in the Yellow River (YR) basin at the end of the 21st century (%).