# Peer review of "Water Scarcity under Various Socio-economic Pathways and its Potential Effects on"

_Hydrology and Earth System Sciences, 2016_

## Referee Comment (RC1) · Anonymous Referee #1 · 12 Jun 2016

General comments

The authors presented a water scarcity assessment of the Yellow River in China using the public database developed by the ISI-MIP project. Since future projections of industrial and domestic water use were not included in the database, the authors estimated them by applying the method proposed by Alcamo et al. (2003) and Flörke et al. (2013). Water scarcity was mainly assessed using the water supply stress index (WaSSI). Future water scarcity was projected to be severe, particularly in the lower stream in late 21st century, mainly due to the growth in industrial water demand.

The Yellow River is widely known as one of the hotspots of water scarcity in the world, hence detailed and comprehensive future water assessments are crucially important.

Although this paper has been excellently prepared as a scientific report, as far as I have observed, the contents are lacking originality and poorly supported by local facts. First, the authors used the WaSSI index. The water scarcity assessment using WaSSI has been established two decades ago by Raskin et al. (1997), Vörösmarty et al. (2000), and Alcamo et al. (2003). Second, the authors used only the output of global hydrological models and highly conceptualized techniques devised for global assessments in this study. I would like to suggest the authors to thoroughly revisit the settings and validate the results of ISI-MIP before using them for local applications. Due to the aforementioned shortcomings, the results and discussion presented in this draft paper are general and not much different from the earlier global water scarcity assessment by Schewe et al. (2014).

Major comments

Line 114 "six global gridded hydrological models": The performance of these models should be validated. In the present form, the authors only showed the mean annual runoff at Lanzhou, Longmen, Sanmenxia, Huayuankou in Supplemental Table S4 without any detailed discussion. At least the reproducibility of monthly river discharge and its inter-annual variations should be assessed. Particularly, Table S4 indicates that the mean annual discharge of MPI-HM and PCR-GLOBWB is approximately half and double of observation in the Yellow River. The rationale of adopting these models in this study must be also clearly described.

Line 121 "The global irrigated and rainfed crop area data (MIRCA2000)": The authors should focus on some of the key simulation settings of ISI-MIP and discuss their validity. For example, ISI-MIP fixed the irrigated and rainfed crop area throughout the 21st century. What are the recent trend in cropland area in this basin? What are the projections by the government and experts? Such local details should be included in this study.

Line 167 "the ratio of human water appropriation (hereafter RHWA)": First, the definition

of this term is missing in the current form of text. The definition and background concept should be clearly stated. Second, the rationale of the thresholds of 50%, 70%, 90% should be carefully discussed. It should be well noted that in many densely populated river basins, total water withdrawal may exceed the total river discharge since treated waste water in upstream is utilized in downstream. Even if the total water withdrawal exceeds the river discharge, water scarcity never occurs if waste water is properly treated and returned to the stream.

Line 181 "The GGCM estimated irrigation water demand": First, the authors should provide the settings and assumptions of this simulation related to water use. What type of crops were planted in the basin in the simulations? Was the crop type varied during the simulations to adopt to warmer climate? Such settings are crucially sensitive to the results. Then carefully discuss whether such simulation conditions are valid for the study basin, and what should be noted in interpreting the results.

Line 195 "Using the historical GDP per capita and industry water use per capita data": Although the authors claimed that their industrial water model followed Alcamo et al. (2003) and Flörke et al. (2013) in line 187, there is a fundamental difference in explanatory variables (input data). In reality, the explanatory variables of Alcamo et al. (2003) and Flörke et al. (2013) were electricity production (a rough indicator of the magnitude of manufacturing output) or value added of industrial sectors respectively, not GDP. In general, industrial water grows much gently than GDP in long term (see Alcamo et al. (2003) and Flörke et al. (2013)). Note that the usage of GDP might be one of the reasons why the industrial water exploded in late 21st century in this study.

Line 200 "In the domestic sector, TC was set as 1% per year": SSP narrates substantially different views of the world (O'Neill et al. 2014). It is a bit odd to me that a same parameter was used for SSP1 (sustainable world) and SSP3 (unsuccessful fragmented world) in this study. For instance, Hanasaki et al. (2013) set different parameter for each SSP to make parameter and narrative scenario consistent.

Line 203 "ratio of water demand to water supply": Define this term more precisely. The terms "water demand" and "water supply" are also unclear.

Line 279 "the WaSSI for total water demand is large than under each SSP, meaning that the water would be scarce at the end of the 21st century": Again, if the water withdrawn in upstream is properly treated upstream and returned back to the stream, water scarcity doesn't occur even if WaSSI exceeds one. Elaborate what are the key problems in the basin in reality, and what can be represented by the WaSSI index.

Line 265 "The water resource shortage is most serious under the conventional development scenario (SSP5)": This is contradictory to the original narrative story line of SSP5 (O'Neill et al., 2014) which depicts a technology-oriented world with high capability of adaptation (humans would control negative consequences of environmental problems by technology). What does "SSP" mean in this study? Is this mean that the authors only took the projections of GDP and population from SSP database?

Minor comments

Line 66 "a grant figure": What is this?

Line 114 "H80", "PRC-GLOBWB": "H08" and "PCR-GLOBWB" respectively

Line 267 "meaning than water demand outstrip supply water": Rephrase this part.

References

Alcamo, J., Döll, P., Henrichs, T., Kaspar, F., Lehner, B., Rösch, T., and Siebert, S.: Development and testing of the WaterGAP 2 global model of water use and availability, Hydrolog. Sci. J., 48, 317-337, 10.1623/hysj.48.3.317.45290, 2003.

Flörke, M., Kynast, E., Bärlund, I., Eisner, S., Wimmer, F., and Alcamo, J.: Domestic and industrial water uses of the past 60 years as a mirror of socio-economic development: A global simulation study, Global Environ. Chang., 23, 144-156, http://dx.doi.org/10.1016/j.gloenvcha.2012.10.018, 2013.

[Figure]

Hanasaki, N., Fujimori, S., Yamamoto, T., Yoshikawa, S., Masaki, Y., Hijioka, Y., Kainuma, M., Kanamori, Y., Masui, T., Takahashi, K., and Kanae, S.: A global water scarcity assessment under Shared Socio-economic Pathways – Part 1: Water use, Hydrol. Earth Syst. Sci., 17, 2375-2391, 10.5194hess-17-2375-2013, 2013.

O'Neill, B., Kriegler, E., Riahi, K., Ebi, K., Hallegatte, S., Carter, T., Mathur, R., and van Vuuren, D.: A new scenario framework for climate change research: the concept of shared socioeconomic pathways, Climatic Change, 122, 387-400, 10.1007/s10584-013-0905-2, 2014.

Raskin, P., Gleick, P., Kirshen, P., Pontius, G., and Strzepek, K.: Comprehensive assessment of the freshwater resources of the world, Stockholm Environment Institute, Stockholm, Sweden, 1997.

Schewe, J., Heinke, J., Gerten, D., Haddeland, I., Arnell, N. W., Clark, D. B., Dankers, R., Eisner, S., Fekete, B. M., Colón-González, F. J., Gosling, S. N., Kim, H., Liu, X., Masaki, Y., Portmann, F. T., Satoh, Y., Stacke, T., Tang, Q., Wada, Y., Wisser, D., Albrecht, T., Frieler, K., Piontek, F., Warszawski, L., and Kabat, P.: Multimodel assessment of water scarcity under climate change, P. Natl. Acad. Sci. USA, 111, 3245-3250, 10.1073/pnas.1222460110, 2014.

Vörösmarty, C. J., Green, P., Salisbury, J., and Lammers, R. B.: Global water resources: Vulnerability from climate change and population growth, Science, 289, 284-288, 2000.

---

## Referee Comment (RC2) · Anonymous Referee #2 · 31 Aug 2016

This manuscript presented an assessment of water scarcity under impacts of climate change and socio-economic development in the Yellow River basin (YR). The authors combined 5 climate models and 6 hydrologic models to obtain 30 climate-hydrology model pairs to estimate water supply from the river. The agricultural water demand is estimated using 30 pairs of climate-crop coupled models (5 climate models × 6 crop models). Industrial and domestic water demands are estimated based on relationships with population and GDP. Based on the ratio of water demand to water supply, the authors concluded that sub-basins in upper reaches of YR will have abundant water, while in middle and lower reaches, water shortage may begin to occur in the next a few decades. The main reason of the water scarcity will be the increase of industrial water

demand.

The manuscript is on a topic of interest to the journal and its findings may have practical values to local managers and residents in YR. The writing may need to be improved. Also, I have doubts about the function of GDP vs. industrial water demand used by the authors, which leads to my doubts about the outcomes of this study. My suggestion would be minor revision.

Specific comments:

1. L139: The full name of "SSP" should be provided before the use of abbreviations (e.g. L136).

2. L162-164: There are only 6 GGHMs right? This 7th GGHM is shown as "GGHM-GCMs" in Table S4. Could the authors provide some explanation about this 7th GGHM?

3. L164-165: Based on Table S4, only WBM has "simulated runoff agrees well with the observed runoff". Maybe add discussion about the performance of different GGHMs and the reasoning of performance differences.

4. L192: It should be "Figure S3 (a)".

5. Figure S3: Typos in x-axis, change "pre" to "per", change "captia" to "capita".

6. L196: As I mentioned earlier. The relationship of GDP and industrial water demand has significant impact on the trend of water demand in the projection period, and therefore it has dominating effect on the outcome of this study. The authors should provide better literature review and methodology explanation about this relationship to further validate their results. One concern I have about this hyperbolic curve is that the range of GDP per capita that the curve is based on, as shown in Figure S3, is not matching with the GDP per capita range in the projection period as shown in Figure S2. After 2025, all the SSPs have GDP per capita greater than 50000 yuan, which is the maximum in Figure S3. As a result, for most part of the projection period, the GDP vs. industrial water demand relationship is at the plateau part of the curve, suggesting a

linear increase of industrial water use with GDP increase. I'm not sure if this is a valid assumption, which leads to my doubts about the study outcome that industrial water demand will be the main contributing factor to water scarcity in the future.

7. L198-202: The effect of technologic advance on water use efficiency is considered in the study as explained here. It seems pretty minimal based on the results. I would suggest to link TC with GDP growth or at least test the sensitivity of industrial water demand to TC.

8. The writing in Section 4.1 and 4.2 needs to be improved. To list a few: L251: Please revise this sentence; L267: Please revise this sentence: L283: Please revise this sentence.

---

## Author Comment (AC1) · 28 Sep 2016

Responses to the Reviewer 1 We truly thank the anonymous reviewer for their constructive comments and suggestions for improving our work. We have addressed all the comments in our revised manuscript. The point-by-point responses to the comments are provided below.

Some general comments:

- Question 1: Although this paper has been excellently prepared as a scientific report, as far as I have observed, the contents are lacking originality and poorly supported by local facts. First, the authors used the WaSSI index. The water scarcity assessment

using WaSSI has been established two decades ago by Raskin et al. (1997), Voros-marty et al. (2000), and Alcamo et al. (2003). Second, the authors used only the output of global hydrological models and highly conceptualized techniques devised for global assessments in this study. I would like to suggest the authors to thoroughly revisit the settings and validate the results of ISI-MIP before using them for local applications. Due to the aforementioned shortcomings, the results and discussion presented in this draft paper are general and not much different from the earlier global water scarcity assessments by Schewe et al. (2014).

- Answer: Thank you for the comments and suggestions. WaSSI is a simple and useful index which considers regional trends in both water supply and demand. Since it was established decades ago, it has been widely used as a metric of water supply stress in many references. So we argue that WaSSI is a proper index to be used. As the water scarcity in Yellow River is largely affected by the changes in both water demand and supply sides, we argue that WaSSI is a proper index to be used. This study differs from Schewe et al. (2014) in several important aspects. Firstly, this study assessed water scarcity at sub-basin scale and considered the water flow regulation rule imple-mented by the local river administration which set limit on water withdrawals for each sub-basin. In contrast, the global study of Schewe et al. (2014) does not consider the regulation rule and cannot assess the effects of water regulation on water stress. Sec-ondly, this study assessed the water stress with the ratio of human water appropriation (RHWA) ranging from 50% to 90% in the Yellow River, which is much higher than the criterion of 40% reduction in discharge that is widely used in the global studies. This localized setting of RHWA enables a more realistic assessment of water scarcity than the global assessment. Thirdly, we have proposed a simple method to correct model simulated water supply. The corrected simulations were evaluated by comparing the ISI-MIP model results against the streamflow observations (see responses to Question 2). Lastly, we further assessed the impacts of water scarcity on agricultural production which was absent in Schewe et al. (2014). As one of the major food production regions in China, the area of cultivated land in the Yellow River basin accounted for 13.3% of

the national totals in the year of 2000. Assessing the potential impacts of water scarcity on agricultural production under a changing environment could help shape adaptation approaches. In the revised version, we have clarified the objective and scientific significance of this study in the introduction section.

- Question 2: Line 114: "six global gridded hydrological models": The performance of these models should be validated. In the present form, the authors only showed the mean annual runoff at Lanzhou, Longmen, Sanmenxia, Huayuankou in Supplemental Table S4 without any detailed discussion. At least the reproducibility of monthly river discharge and its inter-annual discharge of MPI-HM and PCR-GLOBWB is approximately half and double of observation in the Yellow River. The rational of adopting these models in this study must be also clearly described.

- Answer: Thanks for the constructive comments. The global models are usually not calibrated against streamflow observations, and thus often exhibit considerable biases in monthly discharge simulations. However, a recent study showed that the sensitivity of the global models to climate variability is generally comparable to that of the regional models which are calibrated (Hattermann et al., 2016). It suggests the model results, after correction for bias, may be used to assess climate change impacts on water supply. We have proposed a simple method to correct model simulated water supply. The corrected simulations were evaluated with the ISI-MIP models by comparing the model results against the streamflow observations. The results show that the bias-corrected water supply can reproduce well the reference conditions. We have clarified this issue in the revision.

Reference: Hattermann, F. F., Krysanova, V., Gosling, S., Dankers, R., Daggupati, P., Donnelly, C., Flörke, M., Huang, S., Motovilov, Y., Buda, S., Yang, T., Müller, C.,Leng, G., Tang, Q., Portmann, F. T., Hagemann, S., Gerten, D., Wada, Y., Masaki, Y., Alemayehu, T., Satoh, Y., and Samaniego, L., 2016. Cross-scale intercomparison of climate change impacts simulated by regional and global hydrological models in eleven large river basins. Climatic Change, accept.

- Question 3: Line 121: "The global irrigated and rainfed crop area data (MIRCA2000)": The authors should focus on some of the key simulation settings of ISI-MIP and discuss their validity. For example, ISI-MIP fixed the irrigation and rainfed crop area throughout the 21st century. What is the recent trend in cropland area in this basin? What are the projections by the government and experts? Such local details should be included in this study.

- Answer: Thanks for the comments. The cropland area of the Yellow River basin in the 2000s (about 16 million ha estimated by the Ministry of Water Resources of the People's Republic of China, 2013) is quite close to that during the period of 1998-2002 shown in MIRCA2000 (about 16.27 million ha). Although the cropland area may change due to local adaptation to the environmental change, the projection of land use change is beyond the scope of this study. The land use map (i.e. cropland area) is fixed throughout the 21st century in this study. However, the irrigation or rainfed crop area is not fixed. When water shortage occurred (agriculture water availability is not enough for irrigation), we assume the irrigation area would be converted into rainfed. In this way, we can assess the impact of water shortage on agriculture production. We have clarified this in the revised manuscript.

- Question 4: Line 167: "the ratio of human water appropriation (hereafter RHWA)": First, the definition of this term is missing in the current form of text. The definition and background concept should be clearly stated. Second, the rational of the thresholds of 50%, 70%, 90% should be carefully discussed. It should be well noted that in many densely populated river basins, total water withdrawal may exceed the total river discharge since treated waste water in upstream is utilized in downstream. Even if the total water withdrawal exceeds the river discharge, water scarcity never occurs if waste water is properly treated and returned to the stream.

- Answer: Thanks for the comments. In this study, the ratio of human water appropriation (RHWA) describes the fraction of net water withdrawal (Yellow River Conservancy Commission of the Ministry of Water Resources (YRCC), 2013) and is defined as the

annual net water withdrawal divided by the annual runoff. The net water withdrawal is defined as the total water withdrawal minus the water that returns back to the river channel. The threshold values of 50%, 70% and 90% are three different scenarios of human water appropriation. The net water withdrawals of the runoff were occupied 53% during 1980s (Zhang et al., 2004) and 72% presently (YRCC, 2013). If environmental flow requirements in the river basin have greater priority than human society during the period of the study, we assumed that the ratio of human consumptive water appropriation in the basin is 50%. Otherwise, we assumed that the ratio of human consumptive water appropriation in the basin is 90%. 70% is the medium-level scenario. We have added the relevant content in the revision.

Reference: YRCC (Yellow River Conservancy Commission of the Ministry of Water Resources), 2013. Comprehensive planning of Yellow River Basin (2012-2030). Zhengzhou: The Yellow River Water Conservancy Press (in Chinese) Zhang, H. M., Niu, Y. G., Wang, B. X., and Li, S. M., 2004. The Yellow River water resources problems and countermeasures. Hydrology, 24(4), 26-31 (in Chinese)

- Question 5: Line 181: "The GGCM estimated irrigation water demand": First, the authors should provide the setting and assumptions of this simulation related to water use. What types of crops were planted in the basin in the simulations? Was the crop type varied during the simulations to adapt to warmer climate? Such settings are crucially sensitive to the results. Then carefully discuss whether such simulation conditions are valid for the study basin, and what should be noted in interpreting the results.

- Answer: Thanks for the comments. We have added a table (Table S2 in the Supplemental materials) to show the setting of the GGCM. All the GGCMs simulate wheat, maize, and soybean, and all but PEGASUS simulates rice. This study assessed the four crops only. The planting area of these four crops is more than 80% of total crop area in the Yellow River basin. The crop type is fixed during the simulation – no adaptation measures were considered. The main purpose of this study is to assess how

water shortage would affect agricultural production if no adaptation measures were taken. We have added a brief discussion in the revision.

- Question 6: Line 195: "using the historical GDP per capita and industry water use per capita data": Although the authors claimed that their industrial water model followed Alcamo et al. (2003) and Flörke et al. (2013) in line 187, there is a fundamental difference in explanatory variables (input data). In reality, the explanatory variables of Alcamo et al. and Flörke et al. were electricity production (a rough indicator of the magnitude of manufacturing output) or value added of industrial sectors respectively, not GDP. In general, industrial water grows much gently than GDP in long term (see Alcamo et al., 2003 and Flörke et al., 2013). Note that the usage of GDP might be one the reasons why the industrial water exploded in late 21st century in this study.

- Answer: Thanks for the comments. Unfortunately, it is still hard to get the electricity production projection or the value added of manufacturing sectors projection in the Yellow River basin or in China. Alternatively, the GDP projection data over the 21st century is readily available for different socio-economic scenarios. Also, we can obtain the share of manufacturing gross value added in total GDP over the 21st century for OECD and Non-OECD country from the UNEP GEO4 Driver Scenarios (Hughes, 2005). Given that China is a Non-OECD country, we could calculate the value added of manufacturing sector in the 21st century. Based on the industrial net water withdrawal and the calculated value added of manufacturing sector, we have reconstructed the industrial water model followed Flörke et al. (2013) in the revision.

Reference: Hughes, B. B., 2005. UNEP GEO4 diver scenarios (fifth draft). Josef Korbel School of International Studies, University of Denver, Colorado. Flörke, M., Kynast, E., Bärlund, I., Eisner, S., Wimmer, F., and Alcamo, J., 2013. Domestic and industrial water uses of the past 60 years as a mirror of socio-economic development: A global simulation study. Global Environment Change, 23, 144-156, doi: 10.1016/j.gloenvcha.2012.10.018
- Question 7: Line 200: "In the domestic sector, TC was set as 1% per year": SSP narrates substantially different view of the world (O'Neill et al., 2014). It is a bit odd to me that a same parameter was used for SSP1 (sustainable world) and SSP3 (unsuccessful fragmented world) in this study. For instance, Hanasaki et al. (2013) set different parameter for each SSP to make parameter and narrative scenario consistent.

- Answer: Agreed. We have revised the domestic water use estimates following Hanasaki et al. (2013).

- Question 8: Line 203: "ratio of water demand to water supply": Define this term more precisely. The terms "water demand" and "water supply" are also unclear.

- Answer: Thanks for the comments. In this study, the WaSSI was defined as the ratio of annual water demand to annual water supply for a specific watershed. Annual water supply was defined as the total potential surface water available for withdraw from a watershed, and was equal to the annual runoff multiplied by RHWA (the ratio of human water appropriation). Annual water demand represents the sum of net water withdrawals for agricultural, domestic, and industrial uses.

- Question 9: Line 279: "the WaSSI for total water demand is large than 1 under each SSP, meaning that the water would be scare at the end of the 21st century": Again, if the water withdrawn in upstream is properly treated upstream and returned to the stream, water scarcity doesn't occur even if WaSSI exceeds one. Elaborate what are the key problems in the basin in reality, and what can be represented by the WaSSI index.

- Answer: In this study, the WaSSI represents water stress only with respect to net water withdrawals, which is defined as the total water withdrawal minus the water that returns back to the river channel, and measures whether water supplies are sufficient for all net withdrawal requirements within a watershed to be met concurrently. We have added those explanations in the section of Method of the revised manuscript.

[Figure]

- Question 10: Line 265: "The water resource shortage is most serious under the conventional development scenario (SSP5)": This is contradictory to the original narrative story line of SSP5 (O'Neill et al., 2014) which depicts a technology-oriented world with high capability of adaptation (humans would control negative consequences of environmental problems by technology). Water does "SSP" mean in this study? Is this mean that authors only took the projection of GDP and population from SSP database?

- Answer: Thanks for the comments. Each SSP contains a quantitative scenario and a qualitative scenario. The qualitative scenario includes the degree of technological change, overall environmental consciousness and so on. We agree that it is not reasonable to consider GDP and population only. We have taken into account of the effect of technological change and recalculated the water demands following Hanasaki et al. (2013) in the revision.

Reference: Hanasaki, N., Fujimori, S., Yamamoto, T., Yoshikawa, S., Masaki, Y., Hijioka, Y., Kainuma, M., Kanamori, Y., Masui, T., Takahashi, K., and Kanae, S., 2013. A global water scarcity assessment under Shared Socio-economic Pathways – Part 1: Water use. Hydrology and Earth System Sciences, 17, 2375-2391, doi:10.5194/hess-17-2375-2013

Some minor comments:

- Question 11: Line 66 "a grant figure": What is this?

- Answer: Corrected.

- Question 12: Line 114: "H08", "PRC-GLOBWB": "H08" and "PRC-GLOBWB" respectively

- Answer: Revised.

- Question 13: Line 267: "meaning than water demand outstrip supply water": Rephrase this part.

- Answer: Thanks for the comments. In the revision, we have replaced "meaning than water demand outstrip supply water" with "meaning that demand for water outstrips supply".

Please also note the supplement to this comment:
http://www.hydrol-earth-syst-sci-discuss.net/hess-2016-188/hess-2016-188-AC1-supplement.pdf

---

## Author Comment (AC2) · 28 Sep 2016

Responses to the Reviewer 2

We truly thank the anonymous reviewer for their constructive comments and suggestions for improving our work. We have addressed all the comments in our revised manuscript. The point-by-point responses to the comments are provided below.

Some general comments:

- Question 1: The writing may need to be improved.

- Answer: Thanks for the comments. We have carefully polished the language and

grammar thoroughly.

- Question 2: I have doubts about the function of GDP's VS. industrial water demand used by the authors, which leads to my doubts about the outcomes of this study.

- Answer: Thank you for the comments. In this study, the industrial water demand means the industrial net water withdrawal which was defined as the total water withdrawal minus the water that returns back to the river channel. The industrial water demand includes manufacturing water demand and thermoelectric water demand. As we are unable to get the electricity production projection in the Yellow River basin or in China, we assumed that the industrial water demand only include manufacturing water demand in this study. The manufacturing water demand is positively correlated with the economic metric manufacturing gross value added (Dziegielewski et al., 2002). It is more reasonable to estimate industrial water demand with manufacturing gross value added in total GDP than GDP. Based on the obtained GDP projection data and the share of manufacturing gross value added in total GDP over the 21st century from the UNEP GEO4 Driver Scenarios (Hughes, 2005), we have calculated the value added of manufacturing sector from 2010 to 2099. In the revision, we have rebuilt the function of the value added of manufacturing sector and industrial water demand followed Flörke et al. (2013) and recalculated the results.

Reference: Dziegielewski, B., Sharma, S. C., Bik, T. J., Margono, H., and Yang, X., 2002. Analysis of water use trends in the Unites States: 1950-1995. Special Report 28. Illinois Water Resources Center, University of Illinois, USA. Hughes, B. B., 2005. UNEP GEO4 diver scenarios (fifth draft). Josef Korbel School of International Studies, University of Denver, Colorado. Flörke, M., Kynast, E., Bärlund, I., Eisner, S., Wimmer, F., and Alcamo, J., 2013. Domestic and industrial water uses of the past 60 years as a mirror of socio-economic development: A global simulation study. Global Environment Change, 23, 144-156, doi: 10.1016/j.gloenvcha.2012.10.018

Some specific comments:

- Question 3: L139: the full name of "SSP" should be provided before the use of abbreviations (e.g. L136)

- Answer: The full name of "SSPs" is "Shared Socio-economic Pathways". Corrected in the text.

- Question 4: L162-164: There are only 6 GGHMs right? This 7th GGHM is shown as GGHM-GCMs in Table S4. Could the authors provide some explanation about this 7th GGHM?

- Answer: "GGHM-GCMs" in Table S4 is the median of all GGCM-GCMs pairs. We have clarified it in the revision.

- Question 5: L164-165: Based on Table S4, only WBM has "simulated runoff agrees well with the observed runoff". Maybe add discussion about the performance of different GGHMs and the reasoning of performance difference.

- Answer: We have provided the setting and assumptions of the global gridded hydrological models and have added discussion about the performance of different GGHMs. The global models are usually not calibrated against streamflow observation, thus often show a considerable bias in monthly discharge. However, a recent study showed that the sensitivity of the global models to climate variability is in general similar as that of the regional models which are calibrated (Hattermann et al., 2016). It suggests the model results, after correction for bias, may be used to assess climate change impacts on water supply. We have proposed a simple method to correct model simulated water supply. The corrected simulations were evaluated the ISI-MIP models by comparing the model results against the streamflow observations. The results show that the bias-corrected water supply can reproduce well the reference conditions.

Reference: Hattermann, F. F., Krysanova, V., Gosling, S., Dankers, R., Daggupati, P., Donnelly, C., Flörke, M., Huang, S., Motovilov, Y., Buda, S., Yang, T., Müller, C.,Leng, G., Tang, Q., Portmann, F. T., Hagemann, S., Gerten, D., Wada, Y., Masaki, Y., Ale-

mayehu, T., Satoh, Y., and Samaniego, L., 2016. Cross-scale intercomparison of climate change impacts simulated by regional and global hydrological models in eleven large river basins. Climatic Change, accept.

- Question 6: L192: It should be "Figure S3 (a)".

- Answer: Corrected.

- Question 7: Figure S3: Typos in X-axis, change "pre" to "per", change "captia" to "capita"

- Answer: Corrected.

- Question 8: L196: As I mentioned earlier. The relationship of GDP and industrial water demand has significant impact on the trend of water demand in the projection period, and therefore it has dominating effect on the outcome of this study. The authors should provide better literature review and methodology explanation about this relationship to future validate their results. One concern I have about this hyperbolic curve is that the range of GDP per capita that the curve is based on, as shown in Figure S3, is not matching with the GDP per capita range in the projection period as shown in Figure S2. After 2050, all the SSPs have GDP per capita greater than 50000 yuan, which is the maximum in Figure S3. As a result, for most part of the projection period, the GDP vs. industrial water demand relationship is at the plateau part of the curve, suggesting a linear increase of industrial water use with GDP increase. I'm not sure if this is a valid assumption, which leads to my doubts about the study outcome that industrial water demand will be the main contributing factor to water scarcity in the future.

- Answer: Thank you for the comments and suggestions. A number of models have been developed to calculate the industrial water demand quantitatively (e.g. Alcamo, 2003; Hanasaki et al., 2006, 2008; Flörke et al., 2013). Dziegielewski's work (2002) showed that the manufacturing water demand is positively correlated with the economic

metric manufacturing gross value added. We have rebuilt the function of the value added of manufacturing sector and industrial water demand and have recalculated the results (see responses to Question 2).

Reference: Alcamo, J., DoÌLll, P., Henrichs, T., Kaspar, F., Lehner, B., RoÌLsch, T., and Siebert, S., 2003. Development and testing of the WaterGAP 2 global model of water use and availability. Hydrological Sciences Journal, 48, 317-337. Hanasaki, N., Kanae, S., and Oki, T., 2006. A reservoir operation scheme for global river routing models. Journal of Hydrology, 327, 22-41. Hanasaki, N., Kanae, S., Oki, T., Masuda, K., Motoya, K., Shirakawa, N., Shen, Y., and Tanaka, K., 2008. An integrated model for the assessment of global water resources–Part 1: Model description and input meteorological forcing. Hydrology Earth System Sciences, 12, 1007-1025. Dziegielewski, B., Sharma, S. C., Bik, T. J., Margono, H., and Yang, X., 2002. Analysis of water use trends in the Unites States: 1950-1995. Special Report 28. Illinois Water Resources Center, University of Illinois, USA. Flörke, M., Kynast, E., Bärlund, I., Eisner, S., Wimmer, F., and Alcamo, J., 2013. Domestic and industrial water uses of the past 60 years as a mirror of socio-economic development: A global simulation study. Global Environment Change, 23, 144-156, doi: 10.1016/j.gloenvcha.2012.10.018

- Question 9: L198-202: The effect of technologic advance on water use efficiency is considered in the study as explained here. It seems pretty minimal based on the results. I would suggest linking TC with GDP growth or at least test the sensitivity of industrial water demand to TC.

- Answer: Thanks for the suggestion. We have taken into account of the effect of technological change and recalculated the water demands following Hanasaki et al. (2013).

Reference: Hanasaki, N., Fujimori, S., Yamamoto, T., Yoshikawa, S., Masaki, Y., Hijioka, Y., Kainuma, M., Kanamori, Y., Masui, T., Takahashi, K., and Kanae, S., 2013. A global water scarcity assessment under Shared Socio-economic Pathways – Part 1:

Water use. Hydrology and Earth System Sciences, 17, 2375-2391, doi:10.5194/hess-17-2375-2013

- Question 10: The writing in Section 4.1 and 4.2 needs to be improved. To list a few: L251 Please revise this sentence; L267 Please revise this sentence; L283: Please revise this sentence.

- Answer: We have read through the manuscript and improved English writing with help from English editors.

Please also note the supplement to this comment:
http://www.hydrol-earth-syst-sci-discuss.net/hess-2016-188/hess-2016-188-AC2-supplement.pdf

---

## Author Response (AR2)

**Responses to the Reviewer**

We truly thank the editor and the anonymous reviewers for their constructive comments and suggestions for improving our work. We have addressed all the comments in our revised manuscript. The point-by-point responses to the comments are provided below.

**Specific comments:**

- **Question 1:** Line 80: Change "outupts" to "outputs".

**- Answer:** Corrected.

- **Question 2:** Line 92: I suggest change "The mean temperature ranges...." to "The mean annual temperature in 1981-2010 ranges spatially from -5 °C to 15°C...".

**- Answer:** Thanks for the suggestion. We have revised the sentence.

- **Question 3:** Line 94-95: Similar revision suggestion on the mean annual precipitation description.

**- Answer:** Corrected.

- **Question 4:** Line 107: Remove "was".

**- Answer:** Removed.

- **Question 5:** Line 108-110: As shown in Figure S2 (a), the population in the YR basin has a significant decreasing trend in all SSPs. Maybe add some explanation here about this decreasing population projection.

**- Answer:** China's population has been greatly affected by its fertility policy. Under the current fertility policy, many studies have suggested that China's population will continue to grow, and then begin to decrease as the aging of population accelerates (Peng, 2010; Chen and Liu, 2009).

We have added a brief discussion about the population projection in the revision.

**- Answer:** We have rewritten the sentence in the revision.

- **Question 10:** Table 2: In row "Official exchange rate data", column 2, change "country" to

"Country".

**- Answer:** Corrected

- **Question 11:** Figure 4: In column 1, it should be "YR", not "YL" right?

- **Answer:** It should be "YR". Corrected in the revision.

[revised manuscript text omitted]
 before 2025 as both population and domestic water use intensities would increase, and then to decrease due to decrease in population (see Figure S2 (a) in Supplemental material). Industrial water withdrawal is projected to rapidly increase before 2050 as the value added of manufacturing sectors would increase (see Figure S2 (b) in

Supplemental material), and then is projected to decrease slightly due to decrease in industrial water use intensities. The irrigation water withdrawal is projected to decrease from 20 billion m$^3$ yr$^{-1}$ in 1995

to close to 17 billion m$^3$ yr$^{-1}$ in 2084 under RCP 8.5. Irrigation water withdrawal is projected to

4.3 Water abundance/scarcity and sectoral contributions to water scarcity

Figure 5 shows the average annual WaSSI for the YR basin and eight sub-basins throughout the 21st century under the five different SSPs. The WaSSI is projected to increase due to the increase of demand during the 21st century. Under RHWA50, the YR basin is projected to have a WaSSI greater than 1 after 2000s for all SSPs, meaning that water demand outstrips supply. The WaSSI is projected to decrease with the increase of RHWA. Under RHWA70, the water scarcity would not occur in the

21st century for all SSPs. The upper reaches of the YR basin (sub-basins I, II, and III) are projected to have a WaSSI less than 1, meaning that water supply would be more than water demand during the

21st century for all SSPs under all RHWAs. The endorheic basin of the YR basin (sub-basin VIII) is the only region in which the WaSSI is always larger than 1, meaning that the water would be scarce during the 21st century for all SSPs under all RHWAs. In the middle and lower reaches of the YR

basin (sub-basins IV, V, VI, and VII), the WaSSI would begin to be large than 1 at the beginning of the 21st century under RHWA50. With the increase of RHWA, water scarcity would occur later. When the RHWA reaches 70%, water supply would be more than water demand during 1995-2084 in sub- basins IV under all SSPs.

Figure 6 shows the WaSSI calculated as the ratio of annual water demand and sectoral (domestic, industrial and irrigation) water withdrawals to annual water supply under RHWA50 for the YR basin and eight sub-basins at the end of the 21st century under the five different SSPs. In the YR basin, the

WaSSI calculated as annual water demand to water supply is larger than 1 under all SSPs except SSP1, meaning that the water scarcity would occur at the end of the 21st century. Among the three different

[revised manuscript text omitted]

Acknowledgement

This research is supported by the National Natural Science Foundation of China (41425002), the Key Research Program of the Chinese Academy of Sciences (ZDRW-ZS-2016-6-4), and the National Youth Top-notch Talent Support Program in China. We would like to thank anonymous reviewers and the editor for their valuable and constructive suggestions. Thanks are due to Dr. Huijuan Cui for her comments.

[revised manuscript text omitted]